# Factors affecting care of elderly patients among nursing staff at the Ho teaching hospital in Ghana: Implications for geriatric care policy in Ghana

Solomon Mohammed Salia[1]*, Peter Adatara[1], Agani Afaya[1,2], Waliu Salisu Jawula[3], Milipaak Japiong[1], Abubakari Wuni[4], Martin Amogre Ayanore[5], Jacob Erwontaa Bangnidong[6], Felix Hagan[1], Dorcas Sam-Mensah[1], Robert Kaba Alhassan[7]

**1** Department of Nursing, School of Nursing and Midwifery, University of Health and Allied Sciences, Ho, Ghana, **2** College of Nursing, Yonsei University, Yonsei-ro, Seodaemun-gu, Seoul, South Korea, **3** Cambridge Liver Unit, Cambridge University Hospitals NHS Foundation Trust, Addenbrooke's Hospital, Cambridge, United Kingdom, **4** Nurses' and Midwives Training College, Tamale, **5** Department of Health Policy Planning and Management, School of Public Health, University of Health and Allied Sciences, Ho, Ghana, **6** St. Theresa Health Center, Zorkor-Bongo, Bolgatanga, **7** Centre for Health Policy and Implementation Research, Institute of Health Research, University of Health and Allied Sciences, Ho, Ghana

* ssmohammed@uhas.edu.gh, msolomonsalia@gmail.com

**Data Availability Statement:** All relevant data are within the paper and its Supporting Information files.

## Abstract

### Introduction

The population of the aged is increasing globally and in Ghana. In 2020, the population aged over 60 years in Ghana was 2,051,903 and this is expected to reach 2.5 million by 2025 and 6.3 million by 2050. Despite the envisaged increase in the number and life expectancy of the older population in Ghana that will require nursing care, there is a paucity of data on nursing staff knowledge and attitudes toward elderly patients in Ghana.

### Objectives

This study, therefore, assessed factors affecting the care of elderly patients among nursing staff in a tertiary referral health facility in the Volta region of Ghana.

### Methods

The study employed a descriptive cross-sectional design using quantitative data collection approaches. A total of 150 nurses were sampled with a response rate of 95%. Data were analyzed using the Statistical Package for Social Sciences (SPSS) version 23. The analysis included logistic regression to predict factors associated with nurses' knowledge and attitude in caring for elderly patients, after multicollinearity diagnosis and controlling the effect of confounding variables.

### Results

Majority (83.8%) of the nurses demonstrated good knowledge of the aging process, knowledge in the care of the elderly (88.7%), and (84.5%) had a positive caring attitude towards

**Funding:** The authors received no specific funding for this work.

**Competing interests:** The authors have declared that no competing interests exist.

**Abbreviations:** BSc, Bachelor of Science; CI, Confidence Interval; ENT, Ear Nose and Throat; LMICs, Low-and-Middle-Income Countries; N&MC, Nursing and Midwifery Council; OPD, Out Patient Department; OR, Odds Ratio; REC, Research Ethics Committee; GSS, Ghana Statistical Service; UHAS, University of Health and Allied Sciences; UNFPA, United Nations Population Fund; WHO, World Health Organization.

the elderly. Professional education, professional qualification, and knowledge on aged care were significantly associated with nurses' attitude towards the elderly (p<0.001), (p<0.005), and (p<0.010), respectively. Lack of special wards/facilities emerged as the predominantly perceived barrier to caring for the elderly as per the nurses' responses.

## Conclusion

The majority of nurses demonstrated good knowledge and attitude in the aging process and care of the aged. Lack of special wards/facilities and lack of staff motivation were the leading perceived barriers to rendering care to the elderly. Scaling up gerontological nursing programs and establishing special aged care facilities in Ghana with appropriate policy guidelines and regulations for implementation of care will help improve nurses' knowledge and caring attitudes toward the care of elderly patients. Likewise, a national geriatric care policy would help consolidate standard geriatric care in Ghana.

## Introduction

The total population of Ghana was envisaged in 2013 to reach 33.4 million by 2025 [1]. Results from the 2021 Population and Housing Census revealed that the current population of Ghana is 30.8 million people [2] which is envisioned to reach 50 million by 2050 [1]. By 2050, the world's population will reach two billion people above the age of 60, with 400 million aged 80 + years. About 80% of the older population will live in low- and middle-income countries (LMICs) [3].

Additionally, the estimates reveal regional variations of the aged population globally. In 2015, the population above 60 years in Europe was above 27.3% compared to sub-Saharan Africa, which was 5.5%. These figures are projected to increase to 33.6 and 8.3 respectively by 2050 [4]. Ghana's largest population was among children and youth in the past [5], but in recent times, evidence suggests the aged population in the country is on the increase. For instance, in 2020, the population aged 60 and above years was 2,051,903 [6]. This figure is envisaged to reach 2.5 million by 2025 and 6.3 million by 2050 [1]. While the trend suggests all countries are experiencing aging populations, the phenomenon is happening most rapidly in developing countries, including Ghana, where about 60% of the world's older people currently reside [7]. Not only is the aged population increasing rapidly, but there is also evidence that most older people are living longer [5].

Moreover, the impact of this increasing aging population and the longevity of the older people globally may directly impact health care services targeted at the aged, a situation that may affect the attitudes of healthcare providers and aging-related healthcare services [8]. Nurses are said to be the backbone of many healthcare systems providing essential healthcare services to the population. They are also considered one of the many professions providing quality healthcare to the older population, and this role cannot be overlooked [8,9]. Evidence suggests that the quality of care provided by health professionals to the aged is influenced by the perspectives of the healthcare providers [10]. As such, the knowledge and behaviors of nurses can influence their preference for caring for people, which may also affect the type of care they offer.

Globally, several studies have reported on nurses' positive attitudes towards elderly patients during the care delivery process [11–16]. For instance, in Turkey, Polat et al. [16] revealed that

nurses showed respect and patience towards elderly patients due to their age, paid attention to their needs, and prioritized their care over younger patients. Also, in Nigeria, Oyetunde et al. [17] assert that nurses felt good taking care of elderly patients. Similarly, studies in India, [15], and Bangladesh, [18] reported nurses' positive attitudes towards the care of the elderly. Demonstrating their positive attitudes further, the nurses acknowledged the importance of geriatric nursing and recommended the establishment of specialized geriatric clinics and social support systems for older patients.

In contrast, though nurses have shown a positive attitude towards the elderly, several studies have also revealed negative nurses' attitudes regarding the care of elderly patients. Polat et al. [16] revealed that nurses identified elderly patients as weak, disabled, and not flexible. In Nigeria, Oyetunde et al. [17] revealed that caring for older patients is time-consuming and demanding. Also, the study of Kada et al. [19] observed relatively less desirable attitude where the nurses indicated their desire not to become overly attached to dementia patients. In their study, the nurses likened people with dementia to children and that they should not be given life choices.

Several factors have been identified to influence nurses' attitudes towards older patients. In Finland, Eloranta et al. [20] reported that nurses with bachelor's degrees had more positive attitudes towards older patients than those with diploma degrees. Furthermore, post-graduate nurses demonstrated a better caring attitude towards the older patients than bachelor and diploma-educated nurses [21]. Again, Kada et al. [19] identified differences in educational levels to influence nurses' attitudes towards older patients. Their study revealed that auxiliary nurses, ward aids, and nurse assistants were less likely to demonstrate a positive attitude towards the aged than registered nurses. Similarly, other studies reported that nurses who are trained in geriatrics, psychiatric, and dementia care showed more positive attitudes towards older patients than nurses who did not receive any specialized training [19,22–24]. In another study by Liu et al. [21], age was a factor that influenced nurses' attitude towards aged care. Nurses aged between 20 and 30 portrayed a more positive attitude towards older people than those above 30 years, while [16,22] reported no significant association between age and attitude in Turkey and Portugal. The study of Kada et al. [19] also revealed that nurses aged 50 and above portrayed more negative attitudes than nurses younger than 50 years. Additionally, having an extended family type, living with an older person at home, rural residence, and studying in public institutions were also positively associated with nurses' positive attitude towards care of older patients [14,25]. Furthermore, perception about the older patient and age of respondents were associated with attitude in Faronbi et al. [26]. Moreover, adequate knowledge regarding aged care was also associated with positive nurses' attitudes toward the elderly patient [11,27–29].

Several studies have identified barriers affecting nurses' ability to care for elderly patients. Oyetunde et al. [16] in Nigeria have reported that lack of social support, lack of special training programs in gerontology, and lack of special wards/facilities to care for older patients affect older patients' care. Also, Turkish nurses reported a lack of policies regarding geriatric care and disregard for aged care as factors that influence nurses' respect for the older patient during the care delivery process [30].

Empirical evidence suggests an increasing trend in the aged population in Ghana. The growing numbers of the older population and the increase in life expectancy call for an increased need for professional nurses to care for older patients, probably in specialized care centers. There is little or no data regarding nurses' caring attitude towards the older patient in Ghana despite the above indications. There are limited studies identifying the barriers affecting geriatric care in Ghana, especially the Volta region. Therefore, the current study was conducted to assess nurses' attitudes and identify barriers affecting aged care. The study serves as a

stimulus to propel the scaling up of gerontological training programs in nursing by the Nursing and Midwifery Council of Ghana. It also adds to the existing body of knowledge regarding nurses' perspectives of older patient care.

## Materials and methods

### Study design

This study employed a descriptive cross-sectional design using a quantitative approach to data collection to ascertain nurses' attitudes and the barriers affecting the care of the elderly at the Ho Teaching Hospital in the Volta region of Ghana.

### Study area

The study was conducted in the Ho Teaching Hospital, situated in Ho, the administrative capital town of the Volta region. The hospital is the only tertiary-level referral facility in the region. It is approximately a 313-bed capacity referral hospital for the Volta Region and beyond. The hospital has the following wards; Male and Female Medical and Surgical wards, Gynecological ward, Pediatric Ward, Emergency ward, Urology, Trauma ward, Out-Patient Department (OPD), Neonatal Intensive Care Unit, Psychiatric Unit, and Ear, Nose, and Throat (ENT) Unit and Theatre. Data were collected in the following wards/units; medical wards, surgical wards, urology, trauma, OPD, emergency ward, and ENT unit. These wards/units were chosen primarily because they are predominantly managed by nurses whose core responsibility is to render clinical nursing care to patients, including the aged. Paediatric wards and Neonatal Intensive Care Units were excluded because they did not include the care of older patients including maternity, labour and gynecological wards.

### Study population

The study populations were all professional nursing staff (Registered General Nurses) and nonprofessional nursing staff (Registered Nurse Assistants Clinical) who offer direct clinical nursing care to patients, including the older patient at the Ho Teaching Hospital. A total of 220 nurses were eligible to participate in the study in the selected wards/units.

In Ghana, the registered general nurses underwent three years (Diploma) or four years Bachelor of Science (BSc) nursing training program. They are licensed to practice as professional nurses in Ghana after passing Ghana's nursing and Midwifery Council (N &MC) professional licensing exams. As part of the training of the registered general nurses, the curricula used include components of gerontological and home-based nursing by the Nursing and Midwifery Council, Ghana. In addition, the nonprofessional nurses underwent two years of auxiliary training in nursing, either clinical (Registered Nurse Assistant Clinical) or preventive (Registered Nurse Assistant preventive). These categories of nurses serve as nurse's assistants in the clinical or preventive field of practice. Gerontological and home-based nursing is not part of the curriculum for training the registered nurse assistants clinical in Ghana.

### Sample size and sampling determination

Convenience sampling method was used to recruit nurses for the survey. First, the sample size for the study was determined using statistical power analysis. G*Power version 3.1.9.2 guided the sample size calculation [31]. A preliminary power analysis by t-test calculation considering an alpha of 0.05, an effect size of 0.21, and a power of 0.80, the sample size needed using (G-Power 3.1.9.2) was approximately 136. Finally, the sample size for the study was 150 considering a 10% non-response rate.

## Inclusion and exclusion criteria

All registered nurses and registered nurse assistants working in the selected wards who willingly consented to participate in the study were involved in the study. Nurse assistant preventive, nurses undertaking national service, and student nurses who worked in the selected wards were excluded from the study. Additionally, nurses who worked in the selected wards but did not voluntarily consent to participate in the study were excluded from the study. Finally, the study excluded nurses in the selected wards who were on sick leave, study leave or absent from work at the time of data collection.

## Study variables

The main variables of interest were the dependent and independent variables. The main dependent variables of the current study were (knowledge, attitude and barriers). Nurses' knowledge was assessed in the aging process and care of the elderly. Similarly, their attitude towards aged care was assessed. Lastly on the dependent variables is the barriers towards aged care. The independent variables were; Age (≤30, 31–40, 41–50, 51–60); Sex (Male and Female); Marital status (married, single); Religion (Christianity, Muslim and others); Ethnicity (Ewe, Akan, and others); Professional education (Diploma, Bachelor, MPhil and MSc, and others); Professional qualification (Registered nursing and Nurse assistant); Professional rank (staff nursing, senior staff nursing, nursing officer and principal nursing); Nursing specialization (General nurse, Geriatrics and others) and Work experience (1–5, 6–10, 11–15 and more than 15). These variables are discussed in detail in the results section.

## Data collection instrument

A modified standard questionnaire was designed in English to gather nurses' responses on their knowledge and attitudes towards aged care. In designing the questionnaire, the study objectives were considered, and after a careful review of relevant literature on the subject area [13–15,17], questions suitable for the study and relevant to the Ghanaian setting were adapted. The questionnaire was ranked on a five-point Likert scale with appropriate descriptions, thus, "Strongly Disagree," "Disagree," "Neutral," "Agree" and "Strongly Agree" were used. During the analysis stage, the Likert scale was recoded and dichotomized into two groups; Agree (strongly agree and agree) and Disagree (strongly disagree, disagree and neutral). The questionnaire had five main sections. Section 1: nurses demographic characteristics; section 2: nurses' knowledge on the aging process which included five items; section 3: nurses' knowledge towards caring for the elderly with 28 items; section 4: attitude of nurses towards older age with 14 items; section 5: barriers towards the care of the elderly with ten items. Questionnaires were serially numbered to allow for easy identification.

## Data collection procedure

The lead researchers recruited and trained two research assistants who helped in the data collection. Data collection took place between April and May 2018. Questionnaires were self-explanatory and were handed to the nurses individually, which were answered in the nurses' restroom one at a time and were immediately retrieved after completion. This was done at a time suitable to the nurses to avoid any rush in answering the questionnaire. In the wards where the nurses were very busy when the researchers got there, the data collectors rescheduled themselves to a time when the nurses were less busy. Because nurses run different shift systems, time was made available to meet them irrespective of the shift. Each day after data collection, questionnaires were cross-checked in the wards for errors and incompleteness before

taking them home. Also, questionnaires were kept in a sealed envelope for safekeeping by the lead researcher after collection and anonymity and confidentiality of respondent's responses were maintained. After the data collection, missing data were handled using the list-wise deletion of incomplete or missing entries before data analysis. This process ensured that the data set was cleaned for an effective data analysis.

## Validity and reliability

The questionnaire was peer-reviewed by an expert panel involving two adult health nurse specialists and two professors in nursing. The questionnaire was pretested and piloted among 10 nurses in the Ho municipal hospital. These steps were taken to ensure content validity and to determine the suitability of the questionnaire in achieving the study objectives. Cronbach's Alpha coefficient was done to determine the relative internal consistency of the scale and it yielded the following values; knowledge (0.77), attitude (0.83), and barriers (0.93). The Cronbach's Alpha value of our study for attitude is consistent with Arani et al. [32] and Lan and Chen, [33], but contradicts Khagi et al. [24] and Vu et al. [27] where a 0.72 and 0.75 value were reported. Again, our value for knowledge is consistent with Robinson et al. [34], but lower than Amsalu et. Al. [35]. Cronbach's Alpha value of 0.70 are generally acceptable [36], therefore, our value of 0.77 met the minimum requirement and was therefore included in the study.

## Data analysis

Data analysis was done using Statistical Package for Social Sciences (SPSS) version 23. Descriptive statistics such as frequencies, proportions, percentages, means, and standard deviation were used for numerical data. Assessment of nurses' knowledge of the aging process and care of the elderly and attitude towards the elderly were calculated based on percentages. Nurses who performed below the average or average percentage score of less than 50% were classified as having poor knowledge or negative attitude while those who scored above 50% were classified as having good knowledge or a positive attitude for the knowledge and attitude subscales [13,15,17]. Chi-square analysis was performed to determine the association between categorical variables. The significance level of less than 0.05 was considered statistically significant.

Logistic regression was used to determine associations between the dependent and independent variables. Odds ratios, 95% confidence interval, and p-values were calculated using variables that showed significant association (p<0.05) in the chi-square analysis. Assessment of the barriers affecting the care of the elderly was done based on percentages, means, and standard deviation. The percentage of respondents choosing the high response categories, 3–5, was calculated. Their responses were ranked ordered to determine the priority of the barrier items presenting as the top barrier need.

## Ethical approval and consent to participate

The study received ethical approval from the University of Health and Allied Sciences, Research Ethics Committee (REC) (UHAS-REC/A.3[12]17–18). The hospital administration and the nurse managers of the various ward permitted the researchers to collect data in the selected wards. A written informed consent outlining the objectives of the study was obtained from the nurses before the commencement of the study. The nurses were assured of confidentiality, privacy, and anonymity. The nurses were made aware that they had the right to withdraw from the study at any time without any penalty. They were further informed that participation in the survey would involve no direct monetary benefit. However, the findings, when published, will allow for broader reading which may influence positively their overall

clinical practice. The nurses were again informed that the study will pose no risk or discomfort.

## Results

### Demographic characteristics of respondents

Table 1 below shows the demographic characteristics of the respondents. Out of the 150 questionnaires sent to the field, 142 were answered and returned. The findings showed (38.7%) of the nurses were aged 31–40 years. The majority (70.4%) of the nurses were females, Christians (81.0%), and Ewe tribe (53.5%). Regarding professional education, those who received diploma training are the majority (50.0%), while on the professional qualification, the majority (94.4%) were registered general nurses. Regarding professional rankings, senior staff nurses were more (29.6%).

**Table 1. Demographic characteristics of nurses (n = 142).**

| Variable | Frequency (f) | Percent (%) |
|---|---|---|
| **Age of respondents (years)** | | |
| ≤30 | 46 | 32.4 |
| 31–40 | 55 | 38.7 |
| 41–50 | 28 | 19.7 |
| 51–60 | 13 | 9.2 |
| **Sex** | | |
| Male | 42 | 29.6 |
| Female | 100 | 70.4 |
| **Marital status** | | |
| Married | 73 | 51.4 |
| Single | 69 | 48.6 |
| **Religion** | | |
| Christianity | 115 | 81.0 |
| Islam | 13 | 9.2 |
| Others | 14 | 9.8 |
| **Ethnicity** | | |
| Ewe | 76 | 53.5 |
| Akan | 47 | 33.1 |
| Others | 19 | 13.4 |
| **Professional education** | | |
| Diploma in Nursing | 71 | 50.0 |
| Bachelor of Nursing | 54 | 38.0 |
| Msc/MPhil in Nursing | 11 | 7.8 |
| Others | 6 | 4.2 |
| **Professional qualification** | | |
| Registered General Nurse | 134 | 94.4 |
| Enrolled Nurse | 8 | 5.6 |
| **Professional rank** | | |
| Staff Nurse | 18 | 12.7 |
| Senior Staff Nurse | 42 | 29.6 |
| Nursing Officer | 30 | 21.1 |
| Senior Nursing Officer | 33 | 23.2 |
| Principal Nursing Manager | 19 | 13.4 |

*(Continued)*

**Table 1.** (Continued)

| Variable | Frequency (f) | Percent (%) |
|---|---|---|
| **Specialization** | | |
| General Nurse | 116 | 81.7 |
| Geriatrics | 17 | 12.0 |
| Others | 9 | 6.3 |
| **Work experience (years)** | | |
| 1–5 | 33 | 23.2 |
| 6–10 | 65 | 45.8 |
| 11–15 | 20 | 14.1 |
| <15 | 24 | 16.9 |

## Knowledge and attitude towards the care of the elderly

The summary score of the nurses' knowledge and attitude toward the care of the elderly revealed that most nurses showed good knowledge of the aging process (83.8%) while also demonstrating good knowledge towards the care of the elderly (88.7%). Similarly, the majority also portrayed a positive caring attitude towards the elderly (84.5%). The summary of knowledge and attitude is shown in Table 2.

## Barriers towards the care of the elderly

In this study, the lack of special wards/facilities (4.65 ± 0.60) had the highest mean score for barriers affecting the care of the elderly. The second and third highest mean scores were recorded in lack of social support for the elderly (4.64 ± 0.79) and the need for more assistance (4.63 ± 0.87). On the other hand, the least barrier identified in this study was limited literature on geriatric care (3.83 ± 1.42). The percentage of respondents choosing the high response categories (3–5) of the barrier subscale and their means were ranked-ordered to determine the priority presenting barrier items. The three most presenting barriers for consideration were identified as lack of special wards/facilities in the hospital for geriatric care (98.6%), lack of motivation (97.2%), and lack of social support for the elderly (96.5%). The barriers are presented in Table 3.

## Factors that influence nurses' attitude towards the care of the elderly

Table 4 presents logistic regression analysis showing the factors associated with nurses' attitudes towards the elderly in this current study. In a bivariate analysis, the results showed a

**Table 2. Nurses' summary score of knowledge and attitude towards the care of the elderly (n = 142).**

| Variable | Frequency (f) | Percent (%) |
|---|---|---|
| **Knowledge of the aging process** | | |
| Poor knowledge (5–15) | 23 | 16.2 |
| Good knowledge (16–25) | 119 | 83.8 |
| **Knowledge towards elderly care** | | |
| Poor knowledge (28–84) | 16 | 11.3 |
| Good knowledge (85–140) | 126 | 88.7 |
| **Attitude towards elderly patients** | | |
| Negative attitude (14–42) | 22 | 15.5 |
| Positive attitude (43–70) | 120 | 84.5 |

**Table 3. Barriers towards care of the elderly.**

| Variable | Mean Score** ±SD | Response patterns (3–5) f (%) | Priority item rank |
|---|---|---|---|
| Lack of special wards/facilities in the hospitals to care for the elderly | 4.65 ± 0.60 | 140 (98.6) | 1 |
| Lack of motivation in the care of the elderly since it is time-consuming to care for them | 4.50 ± 0.73 | 138 (97.2) | 2 |
| Lack of social support for the elderly | 4.64 ± 0.79 | 137 (96.5) | 3 |
| Patients are likely to need more assistance. | 4.63 ± 0.87 | 135 (95.1) | 4* |
| No clear health care policy for the care of the elderly | 4.33 ± 0.76 | 135 (95.1) | 4* |
| Lack of special training in gerontology to ensure adequate care of the elderly | 4.09 ± 0.98 | 131 (92.3) | 6 |
| Elderly patients exhibit different behaviors which affect their care | 4.08 ± 0.97 | 129 (90.9) | 7 |
| Patients are likely to suffer from more than one ailment | 4.50 ± 1.05 | 128 (90.1) | 8 |
| Lack of interest in studying gerontology | 3.92 ± 1.33 | 119 (83.8) | 9 |
| Limited literature on the care of the elderly | 3.83 ± 1.42 | 109 (76.8) | 10 |

*variables that emerged with the same responses after the responses were rank-ordered.

**Higher mean scores depict a positive knowledge and attitude of nurses in geriatric care and vice-versa.

significant correlation between professional education and attitude towards the elderly (p<0.001). Professional qualification, professional rank, and knowledge also showed a correlation with the nurses' attitude towards the elderly during care (p = 0.005), (p = 0.009), and (p = 0.010), respectively. Furthermore, multivariate logistic regression analysis revealed that nurses who received nonprofessional (nurse assistant) training were 97% times likely to have a positive attitude when caring for the elderly compared to those with diploma training and above [OR = 0.03 (95%CI, 0.00, 0.31), p<0.003]. Consequently, nurse assistant clinical nurses were 84% less likely to demonstrate a positive caring attitude towards the elderly than registered nurses [OR = 0.16 (95%CI, 0.04, 0.68), p = 0.013]. Senior staff nurses, nursing officers, senior nursing officers, and principal nursing officers were 7.6, 5.2, 5.8, and 6.8 times more likely to portray positive attitude towards the older patient during care as compared to staff nurses [OR = 7.60 (95%CI, 1.89, 30.44), p = 0.004], [OR = 5.20 (95%CI, 1.28, 21.18), p = 0.021], [OR = 5.80 (95%CI, 1.43, 23.50), p = 0.014] and [OR = 6.80 (95%CI, 1.19, 38.56), p = 0.030] respectively. Finally, Nurses who demonstrated good knowledge towards the care of the elderly were four times more likely to also portray a good attitude towards the elderly during care than those with insufficient knowledge [OR = 4.13 (95%CI, 1.32, 12.90), p = 0.015]. This is shown in Table 4 below.

## Discussion

This study investigated nursing staff knowledge, attitude, and barriers affecting the care of older patients in a tertiary health facility in Ghana. The findings reveal that 83.8% and 88.7% of the nurses demonstrated good knowledge of the aging process and aged care. Our findings are consistent with earlier studies [18,26,28] where nurses showed good knowledge in the aging process and aged care.

More so, Kabátová et al. [37] in Slovakia showed that the overall knowledge of nurses in the aging process and aged care was good, which agrees with our current findings. However, contrary to our findings, previous studies found significant knowledge gaps regarding the aging process and aged care [23,25,35,38]. For example, as more nurses demonstrated good knowledge in the current study, Vu et al. [27] in Vietnam which assessed knowledge and attitude towards geriatric palliative care among healthcare professionals revealed significant knowledge gaps among the healthcare professionals, where nurses' knowledge score was low (25.8%).

**Table 4. Logistic regression results showing factors associated with nurses' attitude towards the elderly.**

| Variable | Attitude | | Chi-square, χ2 (p-value) | Odds ratio | 95% CI | p-value |
|---|---|---|---|---|---|---|
| | Negative attitude (n = 22) | Positive attitude (n = 120) | | | | |
| **Age of respondents (years)** | | | | | | |
| ≤30 | 8 (36.4) | 38 (31.7) | | Ref. | | |
| 31–40 | 7 (31.8) | 48 (40.0) | | 1.44 | 0.48–4.34 | 0.513 |
| 41–50 | 6 (27.3) | 22 (18.3) | | 0.77 | 0.24–2.52 | 0.668 |
| 51–60 | 1 (4.5) | 12 (10.0) | 1.81 (0.614) | 2.53 | 0.29–22.30 | 0.404 |
| **Sex** | | | | | | |
| Male | 3 (13.6) | 39 (32.5) | | Ref. | | |
| Female | 19(86.4) | 81 (67.5) | 3.18 (0.075) | 0.34 | 0.09–1.17 | 0.087 |
| **Marital status** | | | | | | |
| Married | 12(54.6) | 61 (50.8) | | Ref. | | |
| Single | 10(45.4) | 59 (49.2) | 0.10 (0.749) | 1.16 | 0.47–2.89 | 0.749 |
| **Religion** | | | | | | |
| Christianity | 18(81.8) | 97 (80.8) | | Ref. | | |
| Islam | 3 (13.6) | 10 (8.4) | | 0.62 | 0.15–2.47 | 0.497 |
| Others | 1 (4.6) | 13 (10.8 | 1.32 (0.517) | 2.41 | 0.30–19.61 | 0.410 |
| **Ethnicity** | | | | | | |
| Ewe | 14(63.6) | 62 (51.7) | | Ref. | | |
| Akan | 7 (31.8) | 40 (33.3) | | 1.29 | 0.48–3.47 | 0.614 |
| Others | 1 (4.6) | 18 (15.0) | 2.03 (0.363) | 4.06 | 0.50–33.04 | 0.190 |
| **Professional education** | | | | | | |
| Diploma in Nursing | 10(45.5) | 61 (50.8) | | Ref. | | |
| Bachelor of Nursing | 6 (27.3) | 48 (40.0) | | 1.31 | 0.45–3.86 | 0.623 |
| Msc/MPhil in Nursing | 1 (4.6) | 10 (8.3) | | 1.64 | 0.19–14.24 | 0.654 |
| Nurse Assistant Training | 5 (22.7) | 1 (0.8) | 22.34 (**<0.001**) | 0.03 | 0.00–0.31 | **0.003** |
| Professional qualification | | | | | | |
| Registered General Nurse | 18(81.8) | 116 (96.7) | | Ref. | | |
| Nurse Assistant Clinical | 4 (18.2) | 4 (3.3) | 7.71 (**0.005**) | 0.16 | 0.04–0.68 | **0.013** |
| **Professional rank** | | | | | | |
| Staff Nurse | 8 (36.4) | 10 (8.3) | | Ref. | | |
| Senior Staff Nurse | 4 (18.2) | 38 (31.7) | | 7.60 | 1.89–30.44 | 0.004 |
| Nursing Officer | 4 (18.2) | 26 (21.7) | | 5.20 | 1.28–21.18 | 0.021 |
| Senior Nursing Officer | 4 (18.2) | 29 (24.2) | | 5.80 | 1.43–23.50 | 0.014 |
| Principal Nursing Manager | 2 (9.1) | 17 (14.2) | 13.42 (**0.009**) | 6.80 | 1.19–38.56 | **0.030** |
| **Work experience (years)** | | | | | | |
| 1–5 | 5 (22.7) | 28 (23.3) | | Ref. | | |
| 6–10 | 8 (36.4) | 57 (47.5) | | 1.27 | 0.38–4.24 | 0.695 |
| 11–15 | 4 (18.2) | 16 (13.3) | | 0.71 | 0.17–3.05 | 0.650 |
| <15 | 5 (22.7) | 19 (15.8) | 1.34 (0.720) | 0.68 | 0.17–2.67 | 0.579 |
| **Knowledge of the aging process** | | | | | | |
| Poor | 5 (22.7) | 18 (15.0) | | Ref. | | |
| Good | 17(77.3) | 102 (85.0) | 0.81 (0.366) | 1.67 | 0.55–5.09 | 0.370 |
| **Knowledge of geriatric care** | | | | | | |
| Poor | 6 (27.3) | 10 (8.3) | | Ref. | | |
| Good | 16(72.7) | 110 (91.7) | 6.67 (**0.010**) | 4.13 | 1.32–12.90 | **0.015** |

AOR = Adjusted Odds Ratio; Confidence Intervals were computed at 95% confidence level and p-values less than 0.05 signifies statistical significance.

Nurses may acquire knowledge regarding the care of the aged through experience or education. The nurses need this knowledge to understand the aging process, the complexities associated with older age, and to identify the health problems and needs of the aged to plan appropriate care [39]. Perhaps, the reason for the nurses' good knowledge in this study could be due to the inclusion of gerontological and home-based nursing in the curricula for the training of professional nurses in Ghana. This conclusion confirms the important role education plays in shaping the opinions of nurses about older people [24]. Having good knowledge of both the aging process and aged care enables the nurse who has a vital role in the care delivery process to understand and identify the care needs of older patients with the ultimate aim of rendering quality nursing care [39] and achieving positive patient outcome. Although nurses demonstrated good knowledge regarding the aging process and aged care in this current study, the findings still highlight the importance of gerontological nursing training in Ghana. Gerontological education will further enhance nurses' knowledge to provide quality specialized care to the aged. In support of this point, student nurses in the study of Salin et al. [11] suggested that gerontological nursing should focus on interpersonal skills, medical care, and aging-related diseases to enhance their knowledge and attitude.

This study revealed that the majority (84.5%) of nurses demonstrated positive attitude towards the elderly during the care delivery process, which is consistent with several previous study findings [11,12,14]. The highest positive attitude score (90.9%) was observed when nurses indicated that older patients are cheerful and have a good sense of humor. Also, 88.7% of the nurses felt good taking care of the older patients, while 88.0% mentioned that the older people deserve the care they receive. In Nigeria, Oyetunde et al. [17] similarly reported that nurses generally felt good about their care towards older adults, attended to other clients when caring for older adults, and affirmed that older patients deserve the care nurses offer to them, which all signifies positive attitudes. Similarly, Polat et al. [16] in Turkey revealed that nurses showed respect and patience towards elderly patients, paid attention to their needs, and prioritized caring for them over younger patients during the care delivery process. Another study in India reported that nursing care should be provided to the elderly patients who cannot perform activities of daily living. The study further reported that nurses view older patients as cheerful and people with good humor while also suggesting the need to establish specialized geriatrics clinics and social support systems to care for the elderly patients [15]. Having a positive caring attitude towards the elderly is an essential step to changing the perception of others that older patients are vulnerable and dependent. Whereas a positive nursing attitude is a yardstick to providing more comprehensive and quality nursing care that promotes long life among the elderly, a negative attitude is a recipe for poor quality of care. The negative nursing attitude may prevent older patients from seeking care at health facilities.

Despite an overwhelming positive nurses' attitude in this study, undesirable attitudes were also observed. In particular, nurses indicated that caring for elderly patients was time-consuming, and preferred to give attention to younger patients than older patients. Furthermore, the nurses indicated that older patients are difficult to manage and can easily provoke the caregiver. Skirbekk and Nortvedt [40] assert that nurses and other healthcare providers prioritize caring for younger patients over older ones, acutely ill patients over chronic cases, indicating negative attitudes. It must be emphasized that preferring to care for the younger patients over older ones may amount to discrimination and may affect the quality of care nurses render to the older patient. Polat et al. [16] and Oyetunde et al. [17] assert that nurses view the older patients as weak, disabled, inflexible, and lacking cognitive/mental ability making it difficult to care for them, while some nurses indicated that caring for the elderly patients is time-consuming which confirms our negative attitude findings. More so, in the study of de Almeida Tavares et al. [22], nurses demonstrated poor management of urine incontinence and sleep

disturbances though positive attitudes were also observed in the prevention of pressure injuries and avoidance of restraints. Nonetheless, most of the nurses in the study of Vu et al. [27] showed a neutral attitude, which demonstrated a negative attitude. The abundance of neutral attitudes might be a justification of inadequate knowledge or awareness about aging and aged care, which contradicts our finding.

Multivariate logistic regression analysis to determine the factors that influence nurses' attitude towards older people revealed that knowledge regarding aged care, professional education, and professional qualification were the critical determinants of nurses' attitudes towards the older patient. Specifically, nurses who obtained higher professional nursing training from diploma and above were more likely to demonstrate a positive attitude towards the elderly when providing care than those with education less than a diploma. Consequently, registered general nurses (professional nurses) trained from the diploma level and above portrayed a more positive attitude towards older people than nurse assistants (nonprofessional). Several studies affirm our finding [19,30,41]. The difference in the two categories regarding their attitude level may have been influenced by the curricula used to train nurses in Ghana. The training of professional nurses in Ghana contains gerontological and home-based nursing components which might have influenced attitude towards the care of the aged. Previous studies have reported the important role the nursing curriculum plays in shaping nurses' knowledge and attitude, and subsequently its influence on the work choice [13,24]. As such, the nonprofessional nurses will largely depend on the experiences of their superiors or their instructions to care for the elderly without having any scientific basis, which may affect their knowledge leading to poor attitudes. Similarly, nurses who demonstrated sound knowledge of older age and aged care were more likely to portray a more positive attitude towards the elderly than those with insufficient knowledge. This finding is important as it proves that when people increase their knowledge in aging and aged care their attitude becomes better. This, therefore, necessitates the need for the scaling up of gerontological nursing training in Ghana to meet the envisaged increase in the older population in the future. Holroyd and colleagues [9] agree with our study finding that the most critical factor that influences positive attitude towards care of the older patient is nurses' knowledge about aging and older people. Discrimination among the older people may be associated with inadequate knowledge in geriatrics and gerontology. However, Mellor et al. [41] dispute the idea that lack of knowledge in aging and older people do not necessarily lead to nurses' poor attitude towards older people, rather, nurses' poor knowledge regarding care of the older patient may cause nurses to be unable to perform professional care. Our finding suggests that when nurses know more about older patients, it improves their attitude about the care of older people, and they can render holistic quality care that improves the patient's health. Several previous studies corroborate this finding [11,27,28,42]. Contrary, Afolabi et al. [12] in Nigeria assert that knowledge about older care was not associated with positive nurses' attitude, which may result from the nurses not having any training or previous experience with older people. The studies of [24,43,44] observed that when nurses receive training or education in older care, it has shown to be effective as it advances and improves their knowledge and attitudes about older patient care.

The positive attitude in our study may have also been influenced by the Ghanaian culture that sees most older adults cared for at their family homes where the family lives instead of geriatric homes. In this way, the daily needs and care of the older people are the sole responsibility of the family members. Dignity and respect for older people are core values that family members demonstrate to depict positive attitudes. Also, the practice of the extended family system [14,22], as observed in some parts of Ghana, could have been a reason for the positive attitude towards the older patients though not elicited in the current study. In this family system, younger people have the opportunity to live with their grandparents where they build

good family ties that include the younger ones taking care of the older people, which improves their attitudes. However, with the influence of modernization, urbanization, social media and economic opportunities, more people are drifting away to nuclear family systems which may affect the positive family cohesion and bonding thereby also affecting their attitudes towards older people due to the concentration on one's own family and not the extended family.

Barriers such as lack of special wards/facilities in the hospitals to care for the elderly, lack of motivation in the care of the elderly, lack of social support for the elderly, and lack of clear policy guidelines for the care of the elderly patient were ranked high as factors affecting the care of the elderly in Ghana. Oyetunde et al. [17] reported that hospitals should have specialized units where aged patients can be cared for effectively. Also, nurses in Turkey reported a lack of policies regarding geriatric care and disregard for geriatric patients as factors affecting the care of older patients [30].

## Limitation and strengths

The study was a descriptive cross-sectional study in one single health facility, and the sample size was small, which makes the results not generalizable. Nevertheless, this is one of the few studies conducted in Ghana regarding nursing staff attitudes and factors hindering effective care of the older patients in our healthcare facilities. Thus, the current study creates an opportunity for further extensive research nationally into nurses' knowledge, perception, attitude, and other relevant areas that may affect the aged and their care in Ghana.

## Conclusion

This study revealed that most nurses demonstrated good knowledge and positive attitude towards elderly patients. Lack of special wards/facilities, lack of motivation, lack of social support, and lack of clear policy guidelines to care for the older patients were the top reported barriers by the nurses as factors hindering the care of the elderly. As the aging population is increasing in Ghana, the health care system needs to be improved to meet the aging needs of the people. Improvement in the healthcare systems includes the scaling up of gerontological nursing programs in Ghana to train adequate human resources capable of meeting the health needs of the populace. It also involves establishing special facilities in the various hospitals with appropriate guidelines and regulations, enhancing nurses' knowledge, leading to a positive attitude towards the older patient. We also recommend the collaboration between the Ministry of Health of Ghana and private partners to open special homes for the aged with appropriate care guidelines to enhance the care of the aged. This approach creates access to geriatric care that motivates care providers to deliver appropriate care, which may improve the overall attitude and impact the aged care in healthcare delivery facilities.

## Supporting information

**S1 Data.**
(XLSX)

## Acknowledgments

### Declaration

The authors would like to thank the management of the Ho Teaching Hospital for permitting us the opportunity to conduct our study. We also thank all the nurses who spent precious time participating in our research. We also thank the research assistants for their support.

## Author Contributions

**Conceptualization:** Solomon Mohammed Salia.

**Data curation:** Solomon Mohammed Salia, Peter Adatara, Agani Afaya, Waliu Salisu Jawula, Milipaak Japiong, Abubakari Wuni, Martin Amogre Ayanore, Jacob Erwontaa Bangnidong, Felix Hagan, Dorcas Sam-Mensah, Robert Kaba Alhassan.

**Formal analysis:** Solomon Mohammed Salia, Agani Afaya, Waliu Salisu Jawula, Robert Kaba Alhassan.

**Funding acquisition:** Solomon Mohammed Salia, Peter Adatara, Agani Afaya, Waliu Salisu Jawula, Milipaak Japiong, Abubakari Wuni, Martin Amogre Ayanore, Jacob Erwontaa Bangnidong, Felix Hagan, Dorcas Sam-Mensah, Robert Kaba Alhassan.

**Investigation:** Solomon Mohammed Salia, Peter Adatara, Agani Afaya, Waliu Salisu Jawula, Milipaak Japiong, Abubakari Wuni, Martin Amogre Ayanore, Jacob Erwontaa Bangnidong, Felix Hagan, Dorcas Sam-Mensah, Robert Kaba Alhassan.

**Methodology:** Solomon Mohammed Salia, Peter Adatara, Milipaak Japiong, Martin Amogre Ayanore, Felix Hagan, Dorcas Sam-Mensah, Robert Kaba Alhassan.

**Project administration:** Solomon Mohammed Salia, Peter Adatara, Agani Afaya, Waliu Salisu Jawula, Milipaak Japiong, Abubakari Wuni, Martin Amogre Ayanore, Jacob Erwontaa Bangnidong, Felix Hagan, Dorcas Sam-Mensah, Robert Kaba Alhassan.

**Resources:** Solomon Mohammed Salia.

**Software:** Solomon Mohammed Salia, Peter Adatara, Agani Afaya, Martin Amogre Ayanore, Robert Kaba Alhassan.

**Supervision:** Solomon Mohammed Salia, Peter Adatara.

**Validation:** Solomon Mohammed Salia, Peter Adatara, Martin Amogre Ayanore, Robert Kaba Alhassan.

**Visualization:** Solomon Mohammed Salia, Peter Adatara, Agani Afaya, Waliu Salisu Jawula, Milipaak Japiong, Abubakari Wuni, Martin Amogre Ayanore, Jacob Erwontaa Bangnidong, Felix Hagan, Dorcas Sam-Mensah, Robert Kaba Alhassan.

**Writing – original draft:** Solomon Mohammed Salia.

**Writing – review & editing:** Solomon Mohammed Salia, Peter Adatara, Agani Afaya, Waliu Salisu Jawula, Milipaak Japiong, Abubakari Wuni, Martin Amogre Ayanore, Jacob Erwontaa Bangnidong, Felix Hagan, Dorcas Sam-Mensah, Robert Kaba Alhassan.

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
