## [Decision Letter · Decision Letter 0]

3 Aug 2021

PONE-D-21-14013

Nursing staff Attitudes of and factors affecting care of elderly patients during the care delivery process: a descriptive cross-sectional study.

PLOS ONE

Dear Mr. Salia,

Thank you for submitting your manuscript to PLOS ONE. After careful consideration, we feel that it has merit but does not fully meet PLOS ONE’s publication criteria as it currently stands. Therefore, we invite you to submit a revised version of the manuscript that addresses the points raised during the review process.

Please update the literature review to include recent articles.  Please also address the methodological issues raised by reviewer 1.

We look forward to receiving your revised manuscript.

Kind regards,

Rosemary Frey

Academic Editor

PLOS ONE

Journal Requirements:

Reviewers' comments:

Reviewer's Responses to Questions

**Comments to the Author**

1. Is the manuscript technically sound, and do the data support the conclusions?

Reviewer #1: Yes

Reviewer #2: Yes

2. Has the statistical analysis been performed appropriately and rigorously? 

Reviewer #1: Yes

Reviewer #2: Yes

3. Have the authors made all data underlying the findings in their manuscript fully available?

Reviewer #1: No

Reviewer #2: Yes

4. Is the manuscript presented in an intelligible fashion and written in standard English?

Reviewer #1: No

Reviewer #2: Yes

5. Review Comments to the Author

Reviewer #1: This article is a descriptive cross sectional study regarding nurses’ positive attitudes towards elderly patients and the factors that influence attitudes. The topic is of great significance, given the increase of life expectancy, across the globe and the findings are of particular interest. Thank you for the opportunity to review this manuscript. The tables add greatly to the manuscript. However, the manuscript would be upgraded with a significant editing for writing and language quality. Findings are confirmatory of previous research; however studies which indicated negative attitudes are not cited. A comparison among the different findings (positive VS negative attitudes) would be interesting. Strengths of the manuscript are the well-structured abstract, validity/reliability report and the tables. Weakness of the study is that literature needs to be updated and make a conceptual analysis of the terms that are used.

Major issues

1. The scientific background is explained in the introduction section, though recent research is not cited. This manuscript would be benefited if more recent literature was cited. The originality of the study is supported, however gaps of the literature needs to be addressed. Furthermore, the writers mention the caring attitude, which is a very interesting topic, therefore needs to be developed and put more emphasis how is correlated with the study’s aim.

2. Page 12: Ethical approval: How was nurse’s consent obtained? Verbally? Written?

3. Page 13: Study variables: This paragraph needs better clarification. Maybe a revision would be helpful.

4. Page 13: Data collection: This paragraph needs to be rephrased for better clarification. Where was the instrument development based? Also the use of reference is suggested, since a literature review was made.

5. Page 13: Data collection procedure: How the missing data were handled?

6. Page 14: Data analysis: This paragraph needs further development. How was Likert scale interpreted/calculated? Was it dichotomized? Also, it needs more details on percentages’ calculation.

7. Page 16: Barriers and challenges: The authors should clarify this section to avoid confusion.

8. Page 17: Factors that influence nurses’ attitude towards the care of elderly: This paragraph is quite interesting. The first sentence needs to be rephrased to enhance understanding.

9. Discussion: Study objectives are clearly addressed, but fail to connect findings with previous research in this area. The authors should rewrite the discussion to reference the relevant literature. More specifically:

a) Data description and the importance of the findings are the main strengths of the discussion section. However, the reference of the statistical analysis and a critical analysis of study’s outcome were absence.

b) “Caring attitude”, “empathy” and “compassion” should be developed conceptually. A clear connection with the topic of interests needs to be established.

c) Nurses’ negative attitudes towards elderly have been detected in several studies (Kada et al, 2009, Robinson et al, 2014, Evripidou et al, 2018). This manuscript would be promoted by including those studies and compare the findings.

d) Additionally, factors influencing attitudes needs further development. The article could be enriched by including studies (Elvish et al, 2014, Shinan-Altman et al, 2014, Evripidou et al, 2018) which have made a detailed report to factors that influence nurses’ attitudes.

10. The reference style that the journal uses is Vancouver. Please revise.

Minor issues

11. Page 11: last paragraph, first sentence: needs reference.

12. Study area: It is worth mentioning the reasons for the selection of the particular departments.

13. The validity and reliability paragraph is well structured. The use of reference would be an additional strength of this paragraph.

14. Tables are clear and concise. Table 3 needs to be re-numbered. There are two tables numbered as “Table 4”.

Reviewer #2: Very well done. World elderly population has increased in recent decades and it is foreseen that it will increase in next decades too. So, this type of studies are needed to made world ready to this increase.

The study was well designed to test its hypotesis and well documented. It looks nice and ready to publication.

6. PLOS authors have the option to publish the peer review history of their article (what does this mean?). If published, this will include your full peer review and any attached files.

Reviewer #1: No

Reviewer #2: **Yes: **ismail toygar

---

## [Author Response · Author response to Decision Letter 0]

10 Sep 2021

Manuscript title: 

A descriptive cross-sectional study of factors affecting care of elderly patients among nursing staff at the Ho teaching hospital: implications for geriatric care policy in Ghana

Short title: Factors affecting care of elderly patients among nurses.

*Solomon Mohammed Salia1, Peter Adatara1, Agani Afaya1,2, Waliu Salisu Jawula3, Milipaak Japiong1, Abubakari Wuni4, Martin Amogre Ayanore5, Jacob Erwontaa Bangnidong6, Felix Hagan1, Dorcas Sam-Mensah1, Robert Kaba Alhassan 7

Dear editor, 

We are happy for the opportunity to make inputs into our manuscript to the required journal’s standard for publication. We would like to thank the reviewers for the thorough review and insightful comments on our manuscript. The comments are very useful and we have responded to them to the best of our knowledge. We acknowledge that the comments have no doubt helped improve the quality of our manuscript.

We have therefore responded to the comments to our very best and have provided further details by showing point-by-point feedback on how each of the comments by reviewer # 2 was addressed. For easy identification of our responses, the reviewers’ comments have been repeated while the Authors’ responses appear in BOLD text in the main manuscript.

We also wish to notify you of the change in authorship. The change in authorship have been agreed by all authors to reflect their various contribution to the completion of the manuscript. Peter Adatara is placed as second author from the seventh position while Robert Alhassan Kaba is placed as the last author from the sixth position. We have therefore included a change to authorship form to reflect these changes.

Also, the author’s name “Japion” should be spelt as (Japiong).

Again, the institutional affiliation address of “Waliu Salisu Jawula” has been changed to his new institution’s address.

Additionally, the title of the manuscript has been modified to include the study setting as captured in the revised manuscript

All these changes have appeared on the revised manuscript.

.

Reviewer’s Comments and Authors Responses

Reviewer #1: This article is a descriptive cross sectional study regarding nurses’ positive attitudes towards elderly patients and the factors that influence attitudes. The topic is of great significance, given the increase of life expectancy, across the globe and the findings are of particular interest. Thank you for the opportunity to review this manuscript. The tables add greatly to the manuscript. However, the manuscript would be upgraded with a significant editing for writing and language quality. Findings are confirmatory of previous research; however, studies which indicated negative attitudes are not cited. A comparison among the different findings (positive VS negative attitudes) would be interesting. Strengths of the manuscript are the well-structured abstract, validity/reliability report and the tables. Weakness of the study is that literature needs to be updated and make a conceptual analysis of the terms that are used.

Major issues

Reviewer’s Comment 

1. The scientific background is explained in the introduction section, though recent research is not cited. This manuscript would be benefited if more recent literature was cited. The originality of the study is supported, however gaps of the literature needs to be addressed. Furthermore, the writers mention the caring attitude, which is a very interesting topic, therefore needs to be developed and put more emphasis how is correlated with the study’s aim.

Authors Response

Authors acknowledged these important comments and have therefore revised the background accordingly. Please see the revisions on page 3-4 lines 123-170. 

Reviewer’s Comment

2. Page 12: Ethical approval: How was nurse’s consent obtained? Verbally? Written?

Authors Response

A written informed consent was obtained from the nurses prior to the commencement of the study. This is included in the revised manuscript. Please see page 6 line 237-238.

Reviewer’s Comment

3. Page 13: Study variables: This paragraph needs better clarification. Maybe a revision would be helpful.

Authors Response

Study variables section of the manuscript was revised to include clarification. Please see page 7 line 307-308.

Reviewer’s Comment

4. Page 13: Data collection: This paragraph needs to be rephrased for better clarification. Where was the instrument development based? Also the use of reference is suggested, since a literature review was made.

Authors Response

The data collection instrument section of the manuscript was revised and the changes made in the revised manuscript. Please see the revisions on page 7-8 line 319-332.

Reviewer’s Comment

5. Page 13: Data collection procedure: How the missing data were handled?

Authors Response

Missing data were handled using the list-wise deletion of incomplete/missing entries before data analysis. Please see the revisions on page 8 lines 347-349. 

Reviewer’s Comment

6. Page 14: Data analysis: This paragraph needs further development. How was Likert scale interpreted/calculated? Was it dichotomized? Also, it needs more details on percentages’ calculation.

Authors Response

The questionnaire was ranked on a five-point Likert scale with appropriate descriptions; thus; 1 (strongly disagree), 2 (disagree), 3 (neutral), 4 (agree) and 5 (strongly disagree). During the analysis, the Likert scale was recoded and dichotomized into two groups; Agree (strongly agree and agree) and Disagree (strongly disagree, disagree and neutral). Please see revision on page 8 line 324-326.

In the analysis of knowledge and attitude, nurses’ who obtained less than 50% of the total score were classified as having poor knowledge or negative attitude while those who scored more than 50% of the total score were classified as good knowledge or positive attitude. Please see the revisions on page 9 lines 370-373

Reviewer’s Comment

7. Page 16: Barriers and challenges: The authors should clarify this section to avoid confusion.

Authors Response

The authors have revised the manuscript and have agreed to use the word “barriers” and not challenges. 

Reviewer’s Comment

8. Page 17: Factors that influence nurses’ attitude towards the care of elderly: This paragraph is quite interesting. The first sentence needs to be rephrased to enhance understanding.

Authors Response

The manuscript was revised to rephrase the sentence which added more clarity to the paragraph. Please see the revisions on Page 12 line 457-458. 

Reviewer’s Comment

9. Discussion: Study objectives are clearly addressed but fail to connect findings with previous research in this area. The authors should rewrite the discussion to reference the relevant literature. More specifically:

a) Data description and the importance of the findings are the main strengths of the discussion section. However, the reference of the statistical analysis and a critical analysis of study’s outcome were absence.

b) “Caring attitude”, “empathy” and “compassion” should be developed conceptually. A clear connection with the topic of interests needs to be established.

c) Nurses’ negative attitudes towards elderly have been detected in several studies (Kada et al, 2009, Robinson et al, 2014, Evripidou et al, 2018). This manuscript would be promoted by including those studies and compare the findings.

d) Additionally, factors influencing attitudes needs further development. The article could be enriched by including studies (Elvish et al, 2014, Shinan-Altman et al, 2014, Evripidou et al, 2018) which have made a detailed report to factors that influence nurses’ attitudes.

Authors Response

The authors have revised the discussion section of the manuscript to address all the reviewer comments and have also made inputs based on reviewer suggestions. Please see the revisions on Page 16-19 line 481-628.

Reviewer’s Comment

10. The reference style that the journal uses is Vancouver. Please revise.

Authors Response

The manuscript was revised and the referencing style changed from APA to Vancouver referencing style including intext citations. Please see the revisions on Page 22-24.

Minor issues

11. Page 11: last paragraph, first sentence: needs reference.

Authors Response

This comment has been addressed in page 13 line 476.

Reviewer’s Comment

12. Study area: It is worth mentioning the reasons for the selection of the particular departments. 

Authors Response

The study area section of the manuscript has been revised to include the reason for the choice of the wards for the study. Please see the revisions on Page 6 line 255-260.

Reviewer’s Comment

13. The validity and reliability paragraph is well structured. The use of reference would be an additional strength of this paragraph.

Authors Response

The validity and reliability section of the manuscript was revised to provide references that supported our findings on Cronbach’s Alpha Coefficient values. Please see the revisions on Page 8 line 358-363.

Reviewer’s Comment

14. Tables are clear and concise. Table 3 needs to be re-numbered. There are two tables numbered as “Table 4”.

Authors Response

The authors revised the manuscripts and the table numbers thus; table 3 and table 4 renumbered appropriately. Please see page 12 and 14.

---

## [Decision Letter · Decision Letter 1]

7 Oct 2021

PONE-D-21-14013R1A descriptive cross-sectional study of factors affecting care of elderly patients among nursing staff at the Ho teaching hospital: implications for geriatric care policy in GhanaPLOS ONE

Dear Mr. Salia,

Thank you for submitting your manuscript to PLOS ONE. After careful consideration, we feel that it has merit but does not fully meet PLOS ONE’s publication criteria as it currently stands. Therefore, we invite you to submit a revised version of the manuscript that addresses the points raised during the review.Unfortunately, Reviewer One still has serious concerns regarding your methodology and level of critical analysis in the discussion.  These issues must be addressed before further consideration can be given to your manuscript. Please also review the PLOS ONE guidelines regarding English language quality.  Please seek independent editorial help before submitting your revision. 

We look forward to receiving your revised manuscript.

Kind regards,

Rosemary Frey

Academic Editor

PLOS ONE

Journal Requirements:

Additional Editor Comments (if provided):

Reviewers' comments:

Reviewer's Responses to Questions

**Comments to the Author**

1. If the authors have adequately addressed your comments raised in a previous round of review and you feel that this manuscript is now acceptable for publication, you may indicate that here to bypass the “Comments to the Author” section, enter your conflict of interest statement in the “Confidential to Editor” section, and submit your "Accept" recommendation.

Reviewer #1: (No Response)

Reviewer #2: All comments have been addressed

2. Is the manuscript technically sound, and do the data support the conclusions?

Reviewer #1: Partly

Reviewer #2: Yes

3. Has the statistical analysis been performed appropriately and rigorously? 

Reviewer #1: Yes

Reviewer #2: Yes

4. Have the authors made all data underlying the findings in their manuscript fully available?

Reviewer #1: Yes

Reviewer #2: Yes

5. Is the manuscript presented in an intelligible fashion and written in standard English?

Reviewer #1: No

Reviewer #2: Yes

6. Review Comments to the Author

Reviewer #1: The manuscript has been advanced after the revision. The comments have been adequately responded. Findings support the originality of the study, especially if we consider the lack of data in Ghana. However, the manuscript needs further development in order to be publishable. More particular, a professional language editing would be beneficial. Some paragraphs on the introduction and the discussion needs to be re-written, so as to increase coherence. The discussion has been upgraded after the revision, although lacks of a critical analysis of the study’s findings. The data collection instrument also needs further explanation.

Reviewer #2: Dear Author,

Thank you for your revision, the paper is ready for publication in my opinion.

Best regards.

7. PLOS authors have the option to publish the peer review history of their article (what does this mean?). If published, this will include your full peer review and any attached files.

Reviewer #1: **Yes: **Melina Evripidou

Reviewer #2: **Yes: **ismail toygar

---

## [Author Response · Author response to Decision Letter 1]

26 Nov 2021

Reviewer’s Comment 

A professional language editing would be beneficial. Some paragraphs on the introduction and the discussion needs to be re-written, so as to increase coherence.

Authors Response

Authors acknowledged these important comments and have therefore conducted proof reading on the whole manuscript. Some redundant sentences were removed and, in some cases, rephrased. Some of the rephrased sentences appeared in bold text. The authors also removed some of the references from the citations to prevent a case of “reference over-kill” 

Reviewer’s Comment

The data collection instrument also needs further explanation.

Authors Response

The authors acknowledged this comment as very important. After a careful look at the description of the data collection instrument again all the authors came to a conclusion that the necessary ingredients were all present in its description. Therefore, we did not add further explanation to it.

---

## [Decision Letter · Decision Letter 2]

30 Mar 2022

PONE-D-21-14013R2A descriptive cross-sectional study on factors affecting care of elderly patients among nursing staff at the Ho teaching hospital: implications for geriatric care policy in GhanaPLOS ONE

Dear Mr.Salia,

Thank you for submitting your manuscript to PLOS ONE. After careful consideration, we feel that it has merit but does not fully meet PLOS ONE’s publication criteria as it currently stands. Therefore, we invite you to submit a revised version of the manuscript that addresses the points raised during the review process.

Please revise the manuscript in accordance with the minor definitional revisions requested by Reviewer 3.

We look forward to receiving your revised manuscript.

Kind regards,

Rosemary Frey

Academic Editor

PLOS ONE

Journal Requirements:

Reviewers' comments:

Reviewer's Responses to Questions

**Comments to the Author**

1. If the authors have adequately addressed your comments raised in a previous round of review and you feel that this manuscript is now acceptable for publication, you may indicate that here to bypass the “Comments to the Author” section, enter your conflict of interest statement in the “Confidential to Editor” section, and submit your "Accept" recommendation.

Reviewer #2: All comments have been addressed

Reviewer #3: (No Response)

2. Is the manuscript technically sound, and do the data support the conclusions?

Reviewer #2: Yes

Reviewer #3: Partly

3. Has the statistical analysis been performed appropriately and rigorously? 

Reviewer #2: Yes

Reviewer #3: Yes

4. Have the authors made all data underlying the findings in their manuscript fully available?

Reviewer #2: Yes

Reviewer #3: Yes

5. Is the manuscript presented in an intelligible fashion and written in standard English?

Reviewer #2: Yes

Reviewer #3: Yes

6. Review Comments to the Author

Reviewer #2: (No Response)

Reviewer #3: This is an important and worthy study, with helpful policy implications.

A listing of the questions used to operationalize "good knowledge of the aging process", a "positive caring attitude", "good knowledge of geriatric care" and similar constructs is needed. Or at least cite some exemplary questionnaire items that were used. Otherwise it will not be clear to readers what "good" and "positive" consisted of, and hence will give the appearance of not being sufficiently empirical. This clarification could be done either in the methodology section of the text, or an appendix.

Shorten/condense the article title to something more manageable such as... "Factors affecting care of elderly patients among nursing staff at the Ho teaching hospital in Ghana"

7. PLOS authors have the option to publish the peer review history of their article (what does this mean?). If published, this will include your full peer review and any attached files.

Reviewer #2: **Yes: **ismail toygar

Reviewer #3: No

---

## [Author Response · Author response to Decision Letter 2]

5 Apr 2022

Reviewer’s comment

A listing of the questions used to operationalize "good knowledge of the aging process", a "positive caring attitude", "good knowledge of geriatric care" and similar constructs is needed. Or at least cite some exemplary questionnaire items that were used. Otherwise it will not be clear to readers what "good" and "positive" consisted of, and hence will give the appearance of not being sufficiently empirical. This clarification could be done either in the methodology section of the text, or an appendix

Authors Response

The authors view this comment very useful and believe it will help the readers understand how good and poor knowledge and attitude as used this study meant. This will also add value to the construct of the questionnaire and study. To this, the authors included citations of the studies they modelled when the questionnaire was designed in the “data collection instrument section” of the methodology. Again, the authors have included relevant citations in the “analysis” section of the methodology where the knowledge and attitude classification occurred for further clarity. See in from page 6 line: 256-259 and page 7 line: 306-310.

Reviewer’s comment

Shorten/condense the article title to something more manageable such as... "Factors affecting care of elderly patients among nursing staff at the Ho teaching hospital in Ghana"

Authors Response

The authors find this comment very useful and have shortened the manuscript title to; Factors affecting care of elderly patients among nursing staff at the Ho teaching hospital in Ghana: implications for geriatric care policy in Ghana. See page 1 line 3-4.

---

## [Decision Letter · Decision Letter 3]

12 May 2022

Factors affecting care of elderly patients among nursing staff at the Ho teaching hospital in Ghana: implications for geriatric care policy in Ghana

PONE-D-21-14013R3

Dear Mr. Salia,

We’re pleased to inform you that your manuscript has been judged scientifically suitable for publication and will be formally accepted for publication once it meets all outstanding technical requirements.

Kind regards,

Rosemary Frey

Academic Editor

PLOS ONE

Additional Editor Comments (optional):

Reviewers' comments:

Reviewer's Responses to Questions

**Comments to the Author**

1. If the authors have adequately addressed your comments raised in a previous round of review and you feel that this manuscript is now acceptable for publication, you may indicate that here to bypass the “Comments to the Author” section, enter your conflict of interest statement in the “Confidential to Editor” section, and submit your "Accept" recommendation.

Reviewer #2: All comments have been addressed

Reviewer #3: All comments have been addressed

2. Is the manuscript technically sound, and do the data support the conclusions?

Reviewer #2: Yes

Reviewer #3: Yes

3. Has the statistical analysis been performed appropriately and rigorously? 

Reviewer #2: Yes

Reviewer #3: Yes

4. Have the authors made all data underlying the findings in their manuscript fully available?

Reviewer #2: Yes

Reviewer #3: Yes

5. Is the manuscript presented in an intelligible fashion and written in standard English?

Reviewer #2: Yes

Reviewer #3: Yes

6. Review Comments to the Author

Reviewer #2: The author adressed all recommendations which I mentioned on my previous reviews. I have no more suggestions.

Reviewer #3: (No Response)

7. PLOS authors have the option to publish the peer review history of their article (what does this mean?). If published, this will include your full peer review and any attached files.

Reviewer #2: **Yes: **ismail toygar

Reviewer #3: No

---

## [Editor Report · Acceptance letter]

14 Jun 2022

PONE-D-21-14013R3 

Factors affecting care of elderly patients among nursing staff at the Ho teaching hospital in Ghana: implications for geriatric care policy in Ghana 

Dear Dr. Salia:

I'm pleased to inform you that your manuscript has been deemed suitable for publication in PLOS ONE. Congratulations! Your manuscript is now with our production department. 

Kind regards, 

on behalf of

Dr. Rosemary Frey 

Academic Editor

PLOS ONE